# The Integration of a Three-Dimensional Spheroid Cell Culture Operation in a Circulating Tumor Cell (CTC) Isolation and Purification Process: A Preliminary Study of the Clinical Significance and Prognostic Role of the CTCs Isolated from the Blood Samples of Head and Neck Cancer Patients

**DOI:** 10.3390/cancers11060783

**Published:** 2019-06-06

**Authors:** Chia-Jung Liao, Chia-Hsun Hsieh, Feng-Chun Hung, Hung-Ming Wang, Wen-Pin Chou, Min-Hsien Wu

**Affiliations:** 1Graduate Institute of Biomedical Engineering, Chang Gung University, Taoyuan City 33302, Taiwan; l329735@ms49.hinet.net (C.-J.L.); fjiun@hotmail.com (F.-C.H.); d94522010@ntu.edu.tw (W.-P.C.); 2Division of Haematology/Oncology, Department of Internal Medicine, Chang Gung Memorial Hospital (Linkou), Taoyuan City 33302, Taiwan; wisdom5000@gmail.com (C.-H.H.); whm526@adm.cgmh.org.tw (H.-M.W.); 3Department of Chemical Engineering, Ming Chi University of Technology, New Taipei City 24301, Taiwan

**Keywords:** circulating tumor cells (CTCs), spheroid cell culture, cell isolation and purification, head and neck cancer, epithelial-to-mesenchymal transition (EMT), prognosis

## Abstract

Conventional positive and negative selection-based circulating tumor cell (CTC) isolation methods might generally ignore metastasis-relevant CTCs that underwent epithelial-to- mesenchymal transition and suffer from a low CTC purity problem, respectively. To address these issues, we previously proposed a 2-step CTC isolation method integrating a negative selection CTC isolation and subsequent spheroid cell culture. In addition to its ability to isolate CTCs, more importantly, the spheroid cell culture used could serve as a cell culture model mimicking the process of new tumor tissue formation during cancer metastasis. Therefore, it is promising not only to selectively isolate metastasis-relevant CTCs but also to test the potential of cancer metastasis and thus the prognosis of disease. To explore these issues, experiments were performed. The key findings of this study demonstrated that the method was able to harvest both epithelial (E)- and mesenchymal (M)-type CTCs without selection bias. Moreover, both the M-type CTC count and the information obtained from the multidrug resistance-associated protein 2 (MRP2) and MRP5 gene expression analysis of the CTCs isolated via the 2-step CTC isolation method might be able to serve as prognostic factors for progression-free survival in head and neck squamous cell carcinoma.

## 1. Introduction

Cancer metastasis is a primary cause of cancer-derived death [1]. It is well-recognized that circulating tumor cells (CTCs) are cells shed from primary tumors into adjacent vasculature and subsequently present in the blood circulation of metastatic cancer patients [2]. Reports in the literature have revealed that the counts and molecular characteristics of CTCs are significantly associated with tumor burden, cancer progression, response to anticancer therapies, and patient survival [3]. As a result, fundamental studies on CTCs hold great promise for unraveling the mechanisms behind cancer metastasis, which could facilitate the development of effective therapeutic solutions for future cancer care. Nevertheless, CTCs are extremely rare (e.g., approximately 1–10 cells in 1 × 10^9^ normal blood cells [4]) in whole blood samples, which makes them technically difficult to isolate, purify, and identify.

With recent progress in cell isolation and purification techniques, various CTC isolation methods have been successfully demonstrated, which can be broadly classified as physical and biochemical methods [5]. Overall, the CTC isolation methods based on the physical property differences (e.g., size [6], density [7], or dielectrophoretic force [8]) between the CTCs and the other blood cells are generally believed to be easy to perform and label-free to the harvested cells, but the cell purity of CTCs is less than the purity achieved by the biochemical protocols [9]. Conversely, biochemical-based schemes commonly make use of specific cell surface markers, such as epithelial cell adhesion molecule (EpCAM) and cytokeratins (CKs) to define CTCs. Namely, these CTC isolation and purification techniques use these two surface markers to differentiate the CTCs from the surrounding blood cells [5]. This is primarily based on the fact that EpCAM and CKs are expressed by cancer cells of epithelial origin and are normally absent in normal blood cells. The CTC isolation methods based on this principle are usually referred to as the positive selection of CTCs, which are currently the predominantly adopted methods for CTC isolation and purification [10]. Nevertheless, increasing evidence has revealed that the CTCs in blood circulation are heterogeneous in nature due to the phenomenon of epithelial-to-mesenchymal transition (EMT) [11]. Therefore, the isolation of CTCs based on specific cell markers, such as EpCAM or CKs, could lead to selection bias [12].

EMT, first recognized as a feature of embryogenesis, has been reported to play an important role in cancer progression [13]. EMT allows cancer cells with epithelial characteristics to transform into mesenchymal-type cancer cells, enabling them to acquire migratory ability, environmental stress resistance, and stem cell-like properties [14]. All these cellular features are required for cancer metastasis, recurrence, and the acquisition of drug resistance [15]. The coexistences of epithelial (abbreviated as E)-, mesenchymal (abbreviated as M)-, or biphenotypic-type CTCs have been discovered in the blood samples of metastatic cancer patients [11,16]. Nevertheless, these cancer metastasis-relevant and thus clinically-important CTCs that underwent EMT might be ignored in conventional positive selection-based CTC isolation techniques. This is mainly because these EMT-transformed CTCs could reduce their expression of EpCAM and CKs [17,18]. In addition to the selection bias mentioned above, the CTCs harvested via the conventional positive selection-based CTC isolation schemes are normally labeled with magnetic microbeads or trapped on a functionalized substrate surface, which could limit their use for subsequent CTC-based assays or culture.

To address the abovementioned technical issues, more recently, negative selection-based approaches have been proposed for CTC isolation and purification, by which only blood cells [e.g., CD45^positive (pos)^ or CD235a^pos^ cells] are targeted for depletion using immunomagnetic bead-based cell isolation methods [19,20]. This methodology paves a route to isolate and thus obtain label-free CTCs without the selection bias mentioned previously [12]. However, CTC isolation and purification based on this strategy normally suffer from the technical problem of low cell purity (e.g., less than 4% [9]). This could in turn restrict the use of the harvested CTCs for subsequent analytical works (e.g., gene expression analysis [21,22]). To address the technical hurdles encountered in the positive or negative selection-based CTC isolation methods as aforementioned, we previously proposed a 2-step CTC isolation and purification method that integrated a negative selection-based CTC isolation scheme and a subsequent 8-day spheroid cell culture for the further purification of CTCs [23]. The latter mechanism was mainly based on our findings that the cell viability of leukocytes (i.e., the major cell population in the cell sample obtained after a conventional negative selection-based CTC isolation process) after an 8-day spheroid cell culture was significantly decreased, whereas cancer cells (e.g., CTCs) maintained their viability during the cell culture period [23]. Based on this mechanism, therefore, such a cell culture operation was expected to increase cancer cell purity in the cell cultures. This was further confirmed in our previous clinical tests showing that the average purity of CTC-related cells harvested via the 2-step CTC isolation process was 34.8% [23], which was greatly improved in comparison with that (less than 4%) obtained through conventional negative selection-based CTC isolation schemes [9].

Apart from the function of CTC isolation and purification mentioned above the three-dimensional (3-D) spheroid cell culture used in the 2-step CTC isolation and purification process could also work as a kind of in vitro cell culture model mimicking the process of forming new tumor tissues during distant cancer metastasis [24]. Based on this mechanism, therefore, the 2-step CTC isolation process could be promising for selectively isolating live CTCs that are metastatically and thus clinically meaningful. These CTCs are of great value for subsequent fundamental or applied research. Moreover, the 3-D spheroid cell culture in the 2-step CTC isolation process could also serve as a kind of in vitro assay that could be used to test the potential of cancer metastasis and thus the prognosis of cancer disease. To explore the clinical significance and prognostic roles of the CTCs isolated using the 2-step CTC isolation scheme, experimental studies were carried out in this study. In this work, blood samples obtained from head and neck cancer patients were treated with the 2-step CTC isolation and purification method. This was followed by immunofluorescent staining and the analysis of cancer-related gene expression. The key findings of this study showed that CTC-related cells (i.e., E-CTCs or M-CTCs) were observed in 18 (90.0%) of 20 patients, in which E-CTCs and M-CTCs were observed in 15 (75.0%) and 18 (90.0%) of 20 patients, respectively. These results demonstrated that the 2-step CTC isolation method was able to harvest both E- and M-type CTCs without selection bias. In addition, both M-type CTC counts and the information from the analysis of multidrug resistance-associated protein 2 (MRP2) and MRP5 gene expression of the CTCs harvested via the 2-step CTC isolation protocol were able to serve as prognostic factors of progression-free survival (PFS) in head and neck squamous cell carcinoma (HNSCC). Overall, we have initially explored the clinical significance and prognostic roles of the CTCs isolated via the 2-step CTC isolation method. Further studies with a larger sample size and longer follow-up time are worthy to conduct.

## 2. Materials and Methods

### 2.1. Patient Enrollment

This study was approved by the Institutional Review Board of Chang Gung Memorial Hospital at Linkou, Taiwan (approval IDs: 103-7428B and 104-2595B). Informed consent was obtained from all blood sample donors, and all the methods were carried out in accordance with relevant guidelines. Patients were considered eligible if they (1) had newly diagnosed and untreated head and neck cancer, (2) had histologically or cytopathologically confirmed HNSCC, (3) had disease stages of locally advanced or metastatic, unresectable based on the 8th edition of the American Joint Committee on Cancer criteria, (4) were ≥20 years old, and (5) had adequate liver and renal functions with a sufficient white blood cell (WBC) count to tolerate the anticancer therapies (especially chemotherapy). Patients with synchronous double cancer or previous cancers within the last 5 years were not enrolled. Anticancer treatments consisting of concurrent chemoradiotherapy (CCRT) or chemotherapy alone were given following recent The National Comprehensive Cancer Network (NCCN) guidelines [25] and recommendations from the tumor board committee

### 2.2. Chemotherapy Regimens

Palliative chemotherapy regimens were selected after full discussions with the patient’s family. The regimens mainly included cetuximab (400 mg/m^2^ and then 250 mg/m^2^ weekly for 2–3 consecutive months; Merck, Darmstadt, Germany), cisplatin (50–75 mg/m^2^ biweekly to triweekly), and 5-fluorouracil (700–1000 mg/m^2^/day as a continuous infusion during days 1–4 every 28 days). Other treatments included methotrexate (40 mg/m^2^ or a fixed weekly dose of 50 mg), bleomycin (15 mg weekly), and oral tegafur-uracil (300 mg/m^2^/day; TTY Biopharm Co., Ltd., Taipei, Taiwan). Chemotherapy regimens for chemoradiotherapy mainly included cisplatin (50–75 mg/m^2^ biweekly to triweekly) and 5-fluorouracil (700–1000 mg/m^2^/day as a continuous infusion during days 1–4 every 28 days) or oral tegafur-uracil (300 mg/m^2^/day) every day in a 28-day cycle [26].

### 2.3. Blood Sample Processing Using the 2-Step CTC Isolation and Purification Method

The blood sample for CTC isolation was obtained from cancer patients within 7 days before the first dose of chemotherapy. In the blood drawing process, the first 3–5 mL of blood sample was discarded to prevent epithelial cell contamination. Then, the blood sample obtained was stored in vacutainer tubes with tripotassium ethylenediaminetetraacetic acid (BD Bioscences, San Diego, CA, USA) at 4 °C and soon processed within 4 h. In the subsequent sample processing using the 2-step CTC isolation and purification method [23], briefly, the blood sample was first mixed with erythrocyte lysis buffer (15.5 mM NH_4_Cl, 1.4 mM NaHCO_3_, and 10 μM EDTA, pH 7.3) at a volume ratio of 1:10, followed by a 10-min incubation. After washing twice with phosphate buffered saline (PBS), the cell sample obtained was then processed to deplete leukocytes within the sample using a commercial kit (EasySep Human CD45 Depletion Kit, StemCell Technologies, Vancouver, BC, Canada). All the procedures were carried out according to the manufacturer’s instructions. Briefly, the cell samples were re-suspended in a suspension medium (PBS containing 2% fetal bovine serum and 1 mM EDTA) to form a final cell density of 1 × 10^8^ cells/mL. The Depletion Cocktail was then added into the prepared cell suspension in the ratio of 50 μL reagent per 1 mL of cell suspension, and incubated at RT for 15 min. After that, the well-mixed nanoparticle suspension was added to the treated sample in a ratio of 100 μL reagent per 1 mL of sample. After 10 min incubation, the total volume of abovementioned sample was prepared to 2.5 mL by adding the suspension medium. This was followed by placing the sample-loaded tube into a magnet for 10 min. After that, the cell suspension was obtained using a pipet without disturbing the magnetically-attracted leukocytes on the tube wall. The cell sample obtained was then cultured in a spheroid cell culture model as described previously [23]. Briefly, the cells harvested were prepared in culture medium (RPMI-1640 medium supplemented with 10 ng/mL epidermal growth factor, 10 ng/mL basic fibroblast growth factor, and B-27 supplements; all ingredients were purchased from Gibco, Thermo Fisher Scientific, Inc., Waltham, CA, USA), and then seeded into the wells of a 48-well cell culture microplate (Costar, CORNING, Corning, NY, USA) precoated with 2% (*v*/*v*) agarose (Sigma-Aldrich, Millipore Sigma, St. Louis, MO, USA) for achieving low adhesion purpose. During the 8-day spheroid cell culture, half of the volume of the culture medium was refreshed every 2 days without disturbing the cell clusters formed. In addition, the morphologies of the cultured cells were recorded microscopically on days 0, 4, and 8 of culture. The cell cluster formed after 8 days of culture was defined as microemboli comprising at least two cells.

### 2.4. Immunofluorescence Staining (IF Staining)

After the 2-step CTC isolation and purification process described above, the cell sample of the 8-day culture was harvested. Two-thirds were stained with immunofluorescent dyes to identify and thus quantify the E- and M-type CTCs within the sample. Briefly, the cell sample obtained was equally divided into two parts for quantifying E- and M-type CTCs. Each cell sample was prepared as a thin smear on a glass slide (Smear Gell, GenoStaff Co., Ltd., Tokyo, Japan) [27]. Four percent paraformaldehyde in PBS was used to fix the cells on the slide for 20 min, followed by the permeabilization process using 0.1% Triton X-100 in PBS for 30 min. The treated cell smear sample was then incubated in blocking buffer (0.05% Triton X-100 and 2% bovine serum albumin in PBS) for 30 min [28]. After the abovementioned processes, the cell smear sample was then treated with primary antibodies and incubated for 1 h. After washing the treated cell sample with PBS 2 times, secondary antibodies were added and incubated for another 30 min. The treated cell sample was then washed and finally analyzed using a fluorescence microscope (Nikon, Nikon Instruments, Inc., Melville, NY, USA) for identifying and quantifying the E- and M-type CTCs within the sample. In this study, the antibodies and dyes used were as follows: CD45-PE-conjugated monoclonal antibody (200× dilution; 5B1, MACS, Miltenyi Biotec, Bergisch Gladbach, Germany), CD11b-PE-conjugated monoclonal antibody (100× dilution; M1/70, BD Pharmingen, San Diego, CA, USA), EpCAM monoclonal antibody (500× dilution; clone #028, SinoBiological, Beijing, China), pan-CK polyclonal antibody (500× dilution; Abcam, Cambridge, MA, USA), vimentin (VIM) polyclonal antibody (1000× dilution; GeneTex, Inc., Hsinchu, Taiwan), donkey anti-mouse-Alexa Fluor 488-conjugated antibody (1000× dilution; Invitrogen, Thermo Fisher Scientific, Inc., Waltham, MA, USA), donkey anti-rabbit-Alexa Fluor 488-conjugated antibody (1000× dilution; Invitrogen), and Hoechst 33,342 nuclear dye (5 μg/mL; Invitrogen).

### 2.5. Gene Expression Analysis

In this study, the cell sample after 8 days of spheroid cell culture was also collected for the analysis of cancer-related gene expression [29]. Briefly, one-third of the cell sample was harvested, and its total RNA was extracted using a PicoPure RNA Isolation Kit (Applied Biosystems, Thermo Fisher Scientific, Inc.), followed by cDNA synthesis using a SuperScript IV First-Strand Synthesis System (Thermo Fisher Scientific, Inc.). Then, 10 cycle preamplification of the synthesized cDNA was carried out using TaqMan PreAmp Master Mix (Thermo Fisher Scientific). The cancer-related gene expression of cells was then determined using a TaqMan-based real-time PCR system. The TaqMan assays for each gene tested were purchased from Thermo Fisher Scientific and operated based on the manufacturer’s instructions. β-2-Microglobulin (B2M) was used as the internal control. In this study, the cancer-related genes tested included aldehyde dehydrogenase 1 family member A1 (ALDH1), cadherin 1 (CDH1), CDH2, junction plakoglobin (JUP), keratin 19 (KRT19), MRP1, MRP2, MRP4, MRP5, MRP7, Nanog homeobox (NANOG), octamer-binding transcription factor 3/4 (OCT3/4), prominin 1 (PROM1), snail family transcriptional repressor 1 (SNAI1), SRY-box 2 (SOX2), twist family bHLH transcription factor 1 (TWIST1), and VIM. In this study, the expression levels of target genes relative to the B2M gene were then calculated. Genes with relative expression values ≥ the median value were regarded as the high expression group. Conversely, genes with relative expression values < the median value were considered the low expression group.

### 2.6. Statistical Analysis

In this study, the counts of E- and M-type CTCs, as well as total CTCs in different clinical variable conditions, were compared using the nonparametric Mann-Whitney *U* test (two group comparison) or the nonparametric Kruskal-Wallis test (multiple group comparison) with 2-side significance. The log-rank test was used to compare the survival distributions of two groups. In addition, univariate and multivariate Cox proportional hazards regression analyses under LR forward model [30] were used to evaluate the influence of the clinical variables investigated on the survival of HNSCC patients. Based on the Response Evaluation Criteria in Solid Tumors guidelines (version 1.0), cancer treatment response was classified as complete remission, partial response, stable disease, or progressive disease. Disease-specific progression-free survival (PFS) was calculated from the date of the first CTC sampling to the first instance of cancer-specific disease progression or death from any cause [31]. A *p*-value < 0.05 was regarded as statistically significant. To deal with the probability of multiple tests, the significance *p*-value in the gene study was set at 0.003 by the Bonferroni correction (0.05/n).

## 3. Results and Discussion

### 3.1. The CTCs in the Blood Samples of Head and Neck Cancer Patients Treated with the 2-Step CTC Isolation and Purification Method

Conventional CTC-related studies have centered on the quantification or characterization of the CTCs in blood circulation [32]. It has been reported that approximately 1 × 10^6^ CTCs per gram of the tumor mass are released into the blood circulation daily [33]. Nevertheless, the majority of these CTCs die very soon after entering into the blood circulation [34]. Among these CTCs, only 0.01% reach a distant organ and establish metastasis successfully [35]. This phenomenon addresses the need for a method to selectively harvest these metastasis-relevant CTCs for subsequent assays to obtain information that is more clinically and prognostically meaningful. To address this issue, we previously proposed a 2-step CTC isolation and purification protocol that integrates a negative selection-based CTC isolation scheme and a subsequent 8-day spheroid cell culture process [23]. The latter (3-D spheroid cell culture) is generally regarded as a kind of biomimetic in vitro tumor model [24]. In this study, the purpose of utilizing such a 3-D cell culture operation in the CTC isolation and purification process is that it could mimic the process of forming new 3-D tumor tissues during distant cancer metastasis. Based on this mechanism, therefore, the proposed 2-step CTC isolation and purification protocol could be promising not only to selectively isolate the metastasis-relevant CTCs but also to test the potential of cancer metastasis and thus the prognosis of cancer disease. Nevertheless, the clinical significance, prognostic roles, and properties (e.g., E- or M-type CTCs) of the CTCs obtained via the 2-step CTC isolation process were not investigated in the previous study.

To address the aforementioned issues, a clinical test was carried out in this study. Briefly, twenty patients newly diagnosed with HNSCC [from July 2017 to May 2018 in Chang Gung Memorial Hospital (Linkou)] were enrolled in this study. Blood samples were collected before chemotherapies. The blood samples obtained were then treated with the 2-step CTC isolation and purification, followed by immunofluorescent staining and the analysis of cancer-related gene expression. In the 2-step CTC isolation process, the property (e.g., E- or M-type CTCs) of the CTC-related cells before the spheroid cell culture was not explored due to technical limitation. This is mainly because that the CTC-related cells before the spheroid cell culture (i.e., the CTC-related cells obtained after the negative selection-based CTC isolation method) is low. This makes the analytical work difficult. The overall flow chart of this study is illustrated in Figure 1 (details are described in the Materials and Methods section). Briefly, two-thirds of the cell sample obtained after 8 days of spheroid cell culture was stained with immunofluorescent dyes to identify and thus quantify the E- and M-type CTCs within the sample. In this study, the CD45∪CD11b^negative (neg)^/EpCAM∪CK19^pos^ nucleated cells were defined as E-type CTCs, and the CD45∪CD11b^neg^/VIM^pos^ nucleated cells were defined as M-type CTCs. The cell morphologies and the numbers of each cell type were analyzed microscopically. Figure 2A and B representatively show four immunofluorescence microscopic images for the E- and M-type CTCs, respectively. Overall, CTC-related cells (i.e., E-CTCs or M-CTCs) were observed in 18 (90.0%) of 20 patients. Among them, cell spheres were observed in 2 patients (e.g., the images shown in Figure 2B(k–o)). The cell spheres were composed of CTC-related cells and leukocytes (Figure 2B(k–o)). The interactions of leukocytes (e.g., tumor-associated macrophages and regulatory T cells) with CTCs have also been observed previously [36,37,38,39]. These leukocytes interacting with CTCs are speculated to provide survival advantages for CTCs in the blood stream [36,37,40,41].

In this study, E-CTCs and M-CTCs were observed in 15 (75.0%) and 18 (90.0%) of 20 patients, respectively. As a whole, the mean number of M-CTCs (6.4 ± 7.0) per mL of whole blood was significantly higher than that of E-CTCs (4.2 ± 7.5) (*p* = 0.015, Mann-Whitney *U* test). Reports in the literature have also demonstrated that one CTC might simultaneously express E- and M-type cellular markers [16]. However, these biphenotypic CTCs were not explored in this work because the E-CTCs and M-CTCs were observed separately due to technical limitations. As a whole, the 2-step CTC isolation and purification method was proven to harvest both E- and M-type CTCs without selection bias, as occurred in the conventional positive selection-based CTC isolation schemes [12]. This technical feature is important because both M- and E-type CTCs have been reported to be critical for cancer metastasis [15].

### 3.2. A Preliminary Study of the Clinical Significance of E-CTC and M-CTC Counts in the Blood Samples Treated with the 2-Step CTC Isolation and Purification Process

In recent years, the EMT issue in CTC-relevant studies has been actively explored [42]. Fundamental studies have confirmed that the occurrence of EMT in CTCs is associated with cancer invasion, cancer metastasis, CTC cluster formation, and the genome instability of CTCs [43]. Moreover, it has also been found that the counts of E- or M-type CTCs correlate well with tumor size, cancer stage, or cancer metastasis [44,45]. Furthermore, it was also reported that the counts of E- or M-type CTCs correlate better with the cancer treatment response and the survival of cancer patients in many types of solid tumors than the total CTC count [42,46,47]. Nevertheless, the quantifications of E- and M-type CTCs as well as the total CTCs in these studies were based on the counting of them in the environment of blood circulation. As discussed earlier, most CTCs die soon after entering into the blood circulation. Therefore, the information obtained from these CTCs might not directly link to cancer metastasis. Compared with these studies, the use of 3-D spheroid cell culture in the 2-step CTC isolation protocol could be promising for selectively isolating CTCs that are more metastasis relevant. Based on this fact, the information derived from these CTCs could be more clinically meaningful. To investigate the clinical significance of the counts of E- and M-type CTCs as well as total CTCs in the blood samples of cancer patients treated with the 2-step CTC isolation and purification process, experimental studies were performed. Briefly, the baseline characteristics of the enrolled cancer patients are listed in Table 1. The median follow-up time in this cohort was 6.5 months, ranging from 2.3 to 11.9 months. The counting results of E- and M-type CTCs and total CTCs according to the different variables investigated (e.g., age, gender, tumor site, cancer staging, distant metastasis, disease progression, clusters in cultures, and cancer treatments) are summarized in Table 2.

The mean M-CTC count was higher in the cancer patients with distant metastasis than in those without distant metastasis (*p* = 0.066, approaches the statistical significance of *p* = 0.05) (further experimental works are required to confirm this relationship). This finding could indicate the importance of EMT in cancer metastasis. Further studies on these metastasis-relevant CTCs would be valuable for understanding the mechanism underlying cancer metastasis. Within the experimental conditions investigated, overall, there was no significant difference in the counting results of E- and M-type CTCs and total CTCs under the variables explored in this study except for the following situation: the number of total CTCs (25.3 ± 6.7) was significantly higher in the cancer patients with cell clusters formed after 8-day spheroid culture than (9.0 ± 9.0) in the patients without detectable cell clusters after 8-day spheroid culture (*p* = 0.042, Mann-Whitney *U* test).

### 3.3. Evaluation of the Prognostic Factor of Progression-Free Survival (PFS)

In this preliminary study, we also evaluated the potential prognostic factors of PFS using Cox regression analysis. The variables explored included the baseline clinical features [e.g., age, gender, tumor site, Eastern Cooperative Oncology Group (ECOG) performance status, cancer staging, and cancer treatments] and the counts of E-CTCs, M-CTCs or total CTCs using the 2-step CTC isolation and purification method. In this study, two patients were excluded from the Cox regression analysis because salvage surgeries after CCRT-resulted tumor regression were carried out, which might have greatly altered PFS in this population. In addition, overall survival (OS) was not evaluated in this work because the median survival was not reached until August 2018 (data cutoff date). Among the variables investigated, overall, the results (Table 3) revealed that the M-type CTC counts were the only variable significantly associated with PFS in HNSCC (*p* = 0.029). The multivariate analysis under LR forward model was done, and all the covariates (age, gender, tumor site, ECOG performance status, cancer staging, treatment modality, and CTC counts) in univariate analysis were sent into multivariate analysis. The results showed that M-type CTC counts were the only independent prognostic factor for PFS after adjusted all other covariates (*p* = 0.029, HR = 1.153, 95%CI = 1.015–1.310). This result again highlights the importance of the EMT process in cancer progression [48]. In this study, moreover, the total CTC count could also be associated with PFS in the cancer patients who underwent CCRT or CT [the *p* value (*p* = 0.053) approached the statistical significance of 0.05); Table 3]. Further studies in a large cohort are warranted to confirm the prognostic potential of the CTC number obtained from the 2-step CTC isolation and purification process.

### 3.4. The Relationship Between Gene Expression in the Cells Obtained Via the 2-step CTC Isolation Process and PFS in Cancer Patients

As mentioned above, the 2-step CTC isolation and purification protocol could be promising for selectively isolating metastasis-relevant and thus clinically meaningful CTCs due to the utilization of 3-D spheroid cell culture operation. In addition to correlating the CTC (i.e., E-CTC, M-CTC, and total CTC) counts and clinical outcomes as discussed earlier, this study also investigated gene expression in the CTCs obtained from the 2-step CTC isolation process and their relationship with survival in cancer patients. Briefly, one-third of the cell sample harvested after 8 days of spheroid culture was subjected to gene expression analysis using real-time PCR. In this study, the panel of genes explored included the leukocyte marker (CD45) gene, multiple drug resistance-related genes (MRP1, MRP2, MRP4, MRP5, and MRP7) [49], cancer stem cell (CSC)-related genes (ALDH1, CD133, Nanog, OCT4, and SOX2) [50], and EMT-related genes (CK19, CDH1, CDH2, SNAIL1, TWIST1, VIM, and JUP) [51]. The results showed that the mean relative expression of the CD45 gene (the classical marker for identifying leukocytes) after 8-day spheroid culture was significantly lower than that before the cell culture operation (*p* = 0.002, paired sample *t*-test). Moreover, the expression of the KRT19 gene (one of the classical markers used for identifying CTCs) was detected in 3 of 20 cultures and in 15 of 20 cultures before and after 8 days of spheroid cell culture, respectively. These results again confirmed that the 2-step CTC isolation was able to effectively isolate and purify CTCs, which was in line with our previous findings [23].

Furthermore, the correlations between the gene expression of the cell samples (the results were provided as a Appendix A) after spheroid cell culture and the survival of cancer patients were analyzed using the log-rank test. The Kaplan-Meier survival curves with log-rank *p*-values less than 0.2 are shown in Figure 3A. Among them, low expression of the MRP2 gene and high expression of the MRP5 gene trended to associated with worse survival, with borderline significances of *p* = 0.072 and *p* = 0.061, respectively. Therefore, the risk score of survival was then calculated based on gene expression analysis of the MRP2 and MRP5 genes. Briefly, the patients with low expression of the MRP2 gene or high expression of the MRP5 gene received one point each on the risk score. The sum of the risk scores of the two markers is the final score, which ranges from 0–2. The patients with a final score of 2 were considered the high score group, whereas the others were considered the low score group. The patients were dichotomized according to the risk score, and the survival times of each group were compared. The results (Figure 3B) showed that the cancer patients with high risk scores had worse PFS (*p* = 0.011, log-rank test). This result suggests that considering the information from the two markers (i.e., the gene expression of MRP2 and MRP5) simultaneously (Figure 3B) could increase the power of prognosis compared with the results based on one individual marker (Figure 3A). MRP2 and MRP5 genes belong to the ATP-binding cassette (ABC) transporter family. It has been demonstrated that the expression levels of MRP2 and MRP5 are associated with cell susceptibility to anticancer drugs, such as etoposide, cisplatin, and 5-fluorouracil [52]. A meta-analysis of the Oncomine public datasets (https://www.oncomine.org/resource/main.html) showed that MRP2 is significantly downregulated in breast cancer (1 dataset) and leukemia (3 dataset) compared to normal tissues. The MRP2 expressions in HNSCC were evaluated in 10 dataset. In these evaluations, however, differential expression of MRP2, and the *p*-value are lower than the set threshold (i.e., fold change greater than 2, and *p*-value lower than 0.0001). Therefore, the results of these evaluations had no statistical significance. On the other hand, MRP5 is significantly upregulated in many types of cancer, including breast (6 datasets), cervical (3 datasets), colorectal (1 dataset), esophageal (1 dataset), leukemia (1 dataset), liver (1 dataset), lung (4 datasets), lymphoma (2 datasets), and head and neck cancer (1 dataset), but it is significantly downregulated in brain (1 dataset) and gastric (3 datasets) cancers. Nevertheless, the prognostic roles of MRP2 and MRP5 in head and neck cancer have not yet been well defined and are still controversial [53,54,55,56]. The correlations between MRP2 and MRP5 gene expression and survival of HNSCC patients were analyzed with the Gene Expression Profiling Interactive Analysis (GEPIA, http://gepia.cancer-pku.cn/index.html) web server, which is developed for analyzing the RNA sequencing expression data from the TCGA and the GTEx projects. According to the results from GEPIA, both MRP2 and MRP5 expressions were not significantly associated with survival of HNSCC patients (*n* = 509, patients were dichotomized according to the media expression). However, these results were obtained from tissue blocks instead of CTCs. In this study, overall, we demonstrated that the information obtained from gene expression analysis of the MRP2 and MRP5 genes of the CTCs harvested via the 2-step CTC isolation protocol might be able to serve as a prognostic factor of PFS in HNSCC.

## 4. Conclusions

Growing evidence has revealed that the CTCs in blood circulation are heterogeneous in nature due to the phenomenon of EMT. Therefore, the conventional positive selection-based CTC isolation schemes might ignore metastasis-relevant and thus clinically important CTCs that underwent EMT. To address this issue, negative selection-based approaches have been proposed for CTC isolation, by which only blood cells are targeted for depletion. Nevertheless, CTC isolation based on this strategy normally suffers from the technical problem of low cell purity, which could in turn restrict the use of the harvested CTCs for subsequent analyses. To address the technical hurdles associated with the conventional positive and negative selection-based CTC isolation methods, we proposed a 2-step CTC isolation and purification method that integrates a negative selection-based CTC isolation scheme and a subsequent 8-day spheroid cell culture for the further purification of CTCs. This approach was proven to isolate viable CTCs in a high-purity, label-free manner in our previous study. Apart from the technical features mentioned above, the 3-D spheroid cell culture used in the 2-step CTC isolation process could also function as a kind of in vitro cell culture model mimicking the process of forming new tumor tissue during distant cancer metastasis. As a result, the proposed 2-step CTC isolation protocol could be promising not only to selectively isolate metastasis-relevant CTCs but also to test the potential of cancer metastasis and thus the prognosis of cancer disease. To explore the clinical significance and prognostic roles of the CTCs isolated using the 2-step CTC isolation process, experimental studies were carried out in this work. The key findings of this study showed that CTC-related cells (i.e., E-CTCs and M-CTCs) were observed in 18 (90.0%) of 20 patients, in which E-CTCs and M-CTCs were observed in 15 (75.0%) and 18 (90.0%) of 20 patients, respectively. Based on the results, therefore, the 2-step CTC isolation method was proven to isolate both E- and M-type CTCs without selection bias. In addition, both M-type CTC count and the information obtained from the analysis of MRP2 and MRP5 gene expression of the CTCs harvested via the 2-step CTC isolation protocol proved preliminarily to serve as prognostic factors of progression-free survival in HNSCC. As a whole, we have initially explored the clinical significance and prognostic roles of the CTCs isolated via the 2-step CTC isolation method. Further studies with a larger sample size and longer follow-up time are worthy to conduct. Also, further investigations on the EMT status of the CTC-related cells before and after the spheroid cell culture process will be valuable to reveal the role of spheroid cell culture in the CTC isolation scheme. Moreover, further investigations on the dynamic changes of all gene expressions as tested among all patients will be worthy to carry out to find out their relationships with treatment response, cancer metastasis and patient survival.

## Figures and Tables

**Figure 1 cancers-11-00783-f001:**
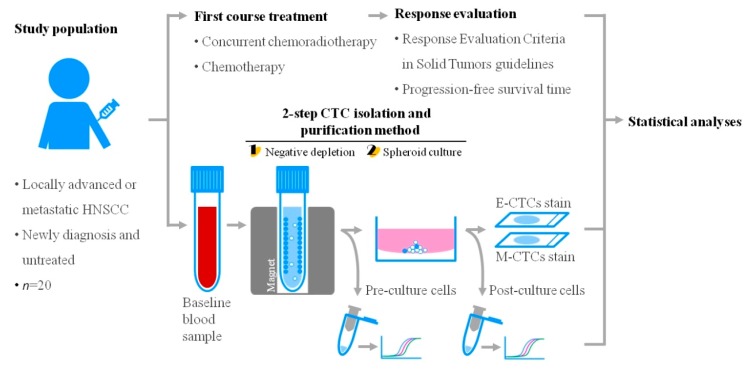
The overall flow chart of this study (HNSCC: head and neck squamous cell carcinoma; E-CTCs: epithelial type-CTCs; M-CTCs: mesenchymal-type CTCs).

**Figure 2 cancers-11-00783-f002:**
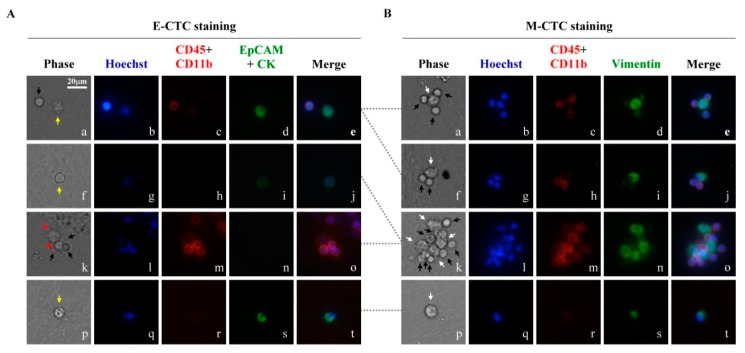
The results of immunofluorescence staining of 8-days cultures. The cultures were divided dichotomously and introduced into E-CTCs staining (**A**) and M-CTCs staining (**B**), respectively. Cells from the same patient were connected with gray dotted-line. Yellow arrows indicate E-CTCs (Hoechst^pos^/CD45∪CD11b^neg^/EpCAM∪CK^pos^ cells); White arrows indicate M-CTCs (Hoechst^pos^/CD45∪CD11b^neg^/VIM^pos^ cells); Black arrows indicate WBCs (Hoechst^pos^/CD45∪CD11b^pos^ cells); Red arrows indicate the cells which were positive for Hoechst staining and negative for the other immunofluorescent dye staining. Scale bar: 20 μm.

**Figure 3 cancers-11-00783-f003:**
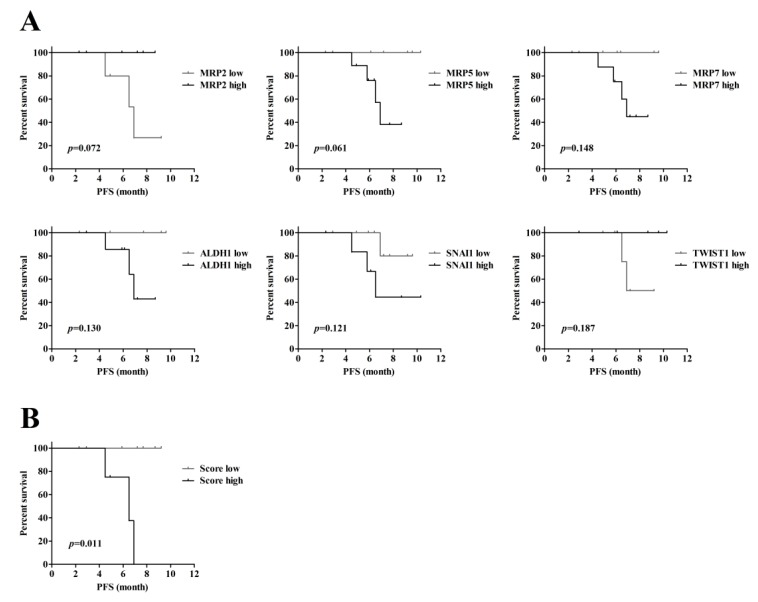
The survival curves of patients who were dichotomized according with the expression of individual cancer-related genes (**A**) or with the risk-score (**B**).

**Table 1 cancers-11-00783-t001:** Patients’ demographic data.

Variable	Category	No	%
Age, year	Median (range)	52.5 (39.5–64.4)	
Gender	Male	17	85
	Female	3	15
Tumor site	Oral cavity	2	10
	Oropharynx	9	45
	Hypopharynx	7	35
	Larynx	2	10
ECOG ^#1^ performance status	0	2	10
	1	14	70
	2	4	20
AJCC stage, 8th edition	II	4	20
	IV	16	80
Distant metastasis	No	15	75
	Yes	5	25
Disease progression	No	13	65
	Yes	5	25
	Missing	2	10
Treatment after blood sample collection	CCRT ^#2^ or CT ^#3^	18	90
	CCRT+ salvage surgery	2	10

^#1^, Eastern Cooperative Oncology Group; ^#2^, Concurrent chemoradiotherapy; ^#3^, Chemotherapy.

**Table 2 cancers-11-00783-t002:** Number of CTCs according to the variable indicated.

Variable	Category	*n*	Post-Culture Cell/mL (Mean ± s.d.)		
			E-CTCs ^#1^	M-CTCs ^#2^	All CTCs ^#3^
Age	<50.0	8	4.7 ± 8.1	8.3 ± 9.2	13.0 ± 11.6
	≥50.0	12	3.9 ± 7.3	5.2 ± 5.2	9.1 ± 8.9
Gender	Male	17	4.7 ± 8.0	7.0 ± 7.5	11.7 ± 10.5
	Female	3	1.5 ± 1.3	3.3 ± 1.1	4.8 ± 2.1
Tumor site	Oral cavity	2	4.2 ± 3.5	9.2 ± 5.2	13.3 ± 1.7
	Oropharynx	9	1.6 ± 1.1	7.6 ± 8.9	9.2 ± 9.1
	Hypopharynx	7	5.2 ± 9.7	4.8 ± 6.1	10.0 ± 11.3
	Larynx	2	12.3 ± 16.5	4.5 ± 2.1	16.8 ± 18.7
AJCC stage, 8th edition	II	4	1.4 ± 1.1	3.0 ± 1.1	4.4 ± 1.9
	IV	16	4.9 ± 8.2	7.3 ± 7.7	12.2 ± 10.6
Distance metastasis	No	15	3.4 ± 6.7	5.3 ± 6.8	8.7 ± 9.3
	Yes	5	6.6 ± 9.8	9.9 ± 7.2 *	16.5 ± 10.6
Treatment after blood sample collection	CCRT ^#4^	16	4.7 ± 8.3	5.3 ± 6.6	10.0 ± 10.4
	CT ^#5^	4	2.3 ± 1.1	10.9 ± 8.0	13.1 ± 8.6
Clusters in cultures	No	18	3.2 ± 6.2	5.9 ± 6.9	9.0 ± 9.0
	Yes	2	13.5 ± 14.8	11.8 ± 8.1	25.3 ± 6.7 **

^#1^, The Hoechst^pos^/CD45∪CD11b^neg^/EpCAM∪CK^pos^ cells in 8-days cultures are defined as E-CTCs; ^#2^, The Hoechst^pos^/CD45∪CD11b^neg^/VIM^pos^ cells in 8-days cultures are defined as M-CTCs; ^#3^, All CTCs was the sum of E-CTCs and M-CTCs in 8-days cultures; ^#4^, Concurrent chemoradiotherapy; ^#5^, Chemotherapy; *, *p* = 0.066; **, *p* < 0.05, Mann-Whitney *U* test.

**Table 3 cancers-11-00783-t003:** Cox regression (univariate) analyses for progression-free survival (PFS).

Variable	HR ^#1^	95% CI ^#2^	*p*
Age	0.875	0.716–1.068	0.189
Gender	0.039	0.000–7309.796	0.601
Tumor site	0.904	0.314–2.599	0.851
ECOG ^#3^ performance status	8.271	0.583–117.370	0.119
AJCC stage, 8th edition	5.633	0.045–698.776	0.482
Treatment after blood collection	0.372	0.073–1.883	0.232
E-CTCs	1.045	0.951–1.147	0.360
M-CTCs	1.153	1.015–1.310	0.029 *
All CTCs	1.101	0.999–1.213	0.053

^#1^, Hazard ratio; ^#2^, Confidence interval; ^#3^, Eastern Cooperative Oncology Group; *, *p*< 0.05.

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
