# Peer review of "The Integration of a Three-Dimensional Spheroid Cell Culture Operation in a Circulating Tumor Cell (CTC) Isolation and Purification Process: A Preliminary Study of the Clinical Significance and Prognostic Role of the CTCs Isolated from the Blood Samples of Head and Neck Cancer Patients"

_cancers, 2019, doi:10.3390/cancers11060783_

Round 1
Reviewer 1 Report
Liao et al. have developed a three-dimensional spheroid two cell culture operation in a circulating tumor cell (CTC) isolation and purification process for identifying metastasis-relevant CTCs that underwent epithelial-to-mesenchymal transition (EMT) and suffered from a low CTC purity problem, respectively. Additionally, they suggested a potential clinical significance and prognostic role of the CTCs isolated from the blood samples of head and neck cancer patients. However, specific concerns about the manuscript are as below:
Major concerns:
- The EMT is a dynamic process during tumor metastasis, especially after colonization. What were changes in the EMT between pre-culture cells and post-culture cells? Also, in vivo xenograft testing but not in vitro TME determine the metastatic potential for the CTCs.
- Please provide the dynamic changes of all gene expressions as tested among all patients. It is unclear whether the gene expressions in CTCs contributed to EMT, stemness, and treatment response as well as tumor metastasis and patient survival, etc.
- It is unclear whether the data is novel for the CTCs isolated via the 2-step CTC isolation method corresponding to the EMT dynamics of CTC and survival. There should be a clear delineation of what is the discovery in the text.
- Low statistical power with few patients and a short follow-up period are unlikely to draw any significant conclusions.
Minor concerns:
- The authors do not sufficiently discuss the dynamic change of CTCs in the EMT.
- It would be informative to show the quantitative PCR analysis for EMT genes and validate the EMT status in these cases.
- In the TCGA dataset, gene expression and survival are available for both MRP2 and MRP5 in patients with HNSCC.
Author Response
Response to reviewer 1’s comments
Liao et al. have developed a three-dimensional spheroid two cell culture operation in a circulating tumor cell (CTC) isolation and purification process for identifying metastasis-relevant CTCs that underwent epithelial-to-mesenchymal transition (EMT) and suffered from a low CTC purity problem, respectively. Additionally, they suggested a potential clinical significance and prognostic role of the CTCs isolated from the blood samples of head and neck cancer patients. However, specific concerns about the manuscript are as below:
Major concerns:
Reviewer’s comment 1: The EMT is a dynamic process during tumor metastasis, especially after colonization. What were changes in the EMT between pre-culture cells and post-culture cells? Also, in vivo xenograft testing but not in vitro TME determine the metastatic potential for the CTCs.
Response 1:
Thanks for reviewer’s insightful comments. We perfectly agree that the EMT is a dynamic process during cancer metastasis. The investigation of the CTC-related cells in terms of their EMT status before and after the 3-D spheroid cell culture (in the proposed 2-step CTC isolation method) is valuable. In this study, however, the dynamic changes of EMT status (e.g., the E- or M-type CTC numbers or gene expression profile) of the CTC-related cells before and after the spheroid cell culture were NOT explored due to technical limitation. This is mainly because that the purity of CTC-related cells before the spheroid cell culture (i.e., the CTC-related cells obtained after the negative selection-based CTC isolation process in the proposed 2-step CTC isolation method) is low (e.g., 4% [Chemical Society reviews 2017, 46:2038]). This phenomenon makes the EMT status evaluation of the CTC-related cells before cell culture technically difficult. Conversely, this problem was solved out after the procedure of the spheroid cell culture (in the proposed 2-step CTC isolation method). This is because that the spheroid cell culture is able to further purify the CTC-related cells (e.g., the average purity of CTC-related cells harvested via the 2-step CTC isolation process: 34.8%). Therefore, this study only investigated the EMT status of the CTC-related cells after the spheroid cell culture. Some of the descriptions/discussions above have now been supplemented in the “Results and discussion” section (Please refer to the manuscript of Highlighted Revision; page 6, lines 258-262).
Regarding the testing model issue raised by the reviewer, again, we agree that the in vivo xenograft testing model could be more biomimetic than the in vitro cell culture model due to its native TME. However, in vitro cell culture-based testing model will be easier to operate than the in vivo xenograft testing model. Spheroid cell culture is the commonly-used in vitro cell culture model for tumor and CTC [Pharmacology & therapeutics 2018, 184:2012; Biotechology Advances 2018, 36:1063]. Therefore, we incorporated the spheroid cell culture in a CTC isolation process to selectively isolate metastasis-relevant CTCs. In addition, we also used the 3-D spheroid cell culture in the 2-step CTC isolation process to test the potential of cancer metastasis and thus the prognosis of cancer disease. Overall, this study just initially explored the feasibility of using a simple in vitro spheroid cell culture in a CTC isolation process to achieve the goals above [Based on the spheroid cell culture model, we found that both the M-type CTC count and the information obtained from the MRP2 and MRP5 gene expression analysis of the CTCs isolated via the 2-step CTC isolation method were able to serve as prognostic factors for progression-free survival in head and neck squamous cell carcinoma (HNSCC)]. We did not specifically investigate the effect of testing model on the prediction of metastatic potential. We thank the reviewer’s insightful comments again.
Reviewer’s comment 2: Please provide the dynamic changes of all gene expressions as tested among all patients. It is unclear whether the gene expressions in CTCs contributed to EMT, stemness, and treatment response as well as tumor metastasis and patient survival, etc.
Response 2:
In current study, single point baseline blood samples were acquired from patients within 7 days before the first course treatment. The dynamic changes of the cancer-related gene expressions of CTCs within an individual patient were not evaluated under current experimental design. For each patient, however, the cancer-related gene expression patterns have been newly provided as supplementary data (Table S1). Also, the prognostic potentials of the expression patterns of each gene were evaluated (Figure 3). However, whether the gene expressions in CTCs contributed to EMT and stemness was not addressed in current study due to insufficient data. This issue is worthy to investigate in the future work. We have highlighted the importance of this issue in the manuscript (Please refer to the manuscript of Highlighted Revision; page 11, lines 390-391; page 13, lines 461-467).
Reviewer’s comment 3: It is unclear whether the data is novel for the CTCs isolated via the 2-step CTC isolation method corresponding to the EMT dynamics of CTC and survival. There should be a clear delineation of what is the discovery in the text.
Response 3:
Thanks for the reviewer’s comments. We proposed the 2-step CTC isolation method for the isolation of label-free, viable, and the all possible CTCs in the blood samples of cancer patients in our previous publication (RSC Advances, 2017, 7:29339). In the previous study, the performance (e.g., CTC purity) of CTC isolation using the proposed method was mainly investigated.
Apart from the function of CTC isolation and purification, the three-dimensional (3-D) spheroid cell culture used in the 2-step CTC isolation and purification process could also work as a kind of in vitro cell culture model mimicking the process of forming new tumor tissues during distant cancer metastasis. As a result, the 2-step CTC isolation process could be promising for selectively isolating live CTCs that are metastatically and thus clinically meaningful. Moreover, the 3-D spheroid cell culture in the 2-step CTC isolation process could also serve as a kind of in vitro assay that could be used to test the potential of cancer metastasis and thus the prognosis of cancer disease. To explore the clinical significance and prognostic roles of the CTCs isolated using the 2-step CTC isolation scheme, experimental studies were carried out in this study. The key findings of this study are that the 2-step CTC isolation method was able to harvest both E- and M-type CTCs without selection bias. In addition, both M-type CTC counts and the information from the analysis of MRP2 and MRP5 gene expression of the CTCs harvested via the 2-step CTC isolation protocol were able to serve as prognostic factors of progression-free survival (PFS) in head and neck squamous cell carcinoma (HNSCC).
All the above-mentioned descriptions were in the original manuscript. In order to clearly highlight the new findings in this study, some of the descriptions in the manuscript have now been modified (Please refer to the manuscript of Highlighted Revision; page 1, line 31; page 3, lines 113 and 116; page 13, lines 453-454).
Reviewer’s comment 4: Low statistical power with few patients and a short follow-up period are unlikely to draw any significant conclusions.
Response 4
Many thanks for the reviewer’s comments. We agree that small sample size and short follow-up period are the weakness of this study. In this work, we have initially explored the clinical significances of the CTCs isolated via the 2-step CTC isolation method. Further studies with larger sample size and longer follow-up time are worthy to conduct. The above descriptions have now been supplemented in the “Introduction”, and “Conclusions” section (Please refer to the manuscript of Highlighted Revision; page 3, lines 123-125; page 13, lines 460-467).
Minor concerns:
Reviewer’s comment 5: The authors do not sufficiently discuss the dynamic change of CTCs in the EMT.
Response 5:
We thank for the comments. As mentioned in response 1, we did NOT explore the dynamic changes of CTCs in terms of EMT due to the technical problem of low CTC purity before the spheroid cell culture operation. Reviewer’s understanding is greatly appreciated. In addition, the major aim of this study is to explore the clinical significance and prognostic role of the CTCs isolated from the blood samples of head and neck cancer patients using the proposed 2-step CTC isolation method. We perfectly agree that the investigation of the CTC-related cells in terms of their EMT status before and after the 3-D spheroid cell culture (in the proposed 2-step CTC isolation method) is valuable. We have highlighted the importance of exploring this issue in the manuscript (Please refer to the manuscript of Highlighted Revision; page 13, lines 460-467).
Reviewer’s comment 6: It would be informative to show the quantitative PCR analysis for EMT genes and validate the EMT status in these cases.
Response 6:
We thank for the valuable comments. Min Yu and coworkers (Science, 2013, 339:580) have demonstrated an association of mesenchymal CTCs with disease progression by performing a serial CTC monitoring in 11 breast cancer patients. Their study confirmed the role of EMT in metastasis of breast cancer. Apart from correlating the CTC (i.e., E-CTC, M-CTC, and total CTC) counts and clinical outcomes, we also investigated gene expression in the CTCs obtained from the 2-step CTC isolation process and their relationship with survival in cancer patients in this study. The panel of genes explored included the leukocyte marker (CD45) gene, multiple drug resistance-related genes (MRP1, MRP2, MRP4, MRP5, and MRP7) [49], cancer stem cell (CSC)-related genes (ALDH1, CD133, Nanog, OCT4, and SOX2) [50], and EMT-related genes (CK19, CDH1, CDH2, SNAIL1, TWIST1, VIM, and JUP). Within the experimental conditions investigated, however, the resuls of EMT-related gene expression analysis showed no stastical significance with the survival in cancer patients. Further studies with larger sample size and longer follow-up time are required to confirm the relationship bweteen the EMT-related gene expression or EMT status and clinical outcomes.
Reviewer’s comment 7: In the TCGA dataset, gene expression and survival are available for both MRP2 and MRP5 in patients with HNSCC.
Response 7:
We thank for reminding us this issue. We are sorry for the imprecise descriptions about the meta-analysis of MRP2 gene in the original manuscript. The MRP2 expressions in HNSCC were evaluated in 10 dataset, which had been recorded in the Omcomine database. In these 10 dataset, however, differential expression of MRP2, and the p-value are lower than the set threshold (i.e., fold change greater than 2, and p-value lower than 0.0001). Therefore, the results of these evaluations had no statistical significance. The original descriptions have now been corrected (Please refer to the manuscript of Highlighted Revision; page 11, lines 412-416). Besides, according to the reviewer’s suggestion, the correlations between MRP2 and MRP5 gene expression and survival of HNSCC patients have been analyzed with the Gene Expression Profiling Interactive Analysis (GEPIA) web server, which is developed for analyzing the RNA sequencing expression data from the TCGA and the GTEx projects. According to the results from GEPIA, both MRP2 and MRP5 expressions were not significantly associated with survival of HNSCC patients (n=509, patients were dichotomized according to the media expression). However, these results were obtained from tissue blocks instead of CTCs. The description above has been added to the Result and discussion section. (please refer to the manuscript of Highlighted Revision; page 12, lines 422-429)

Reviewer 2 Report
This is a very interesting article, looking at the culture and isolation of circulating tumor cells in a head and neck cancer population. The authors look at the growth of the cells, the relative frequency of epithelial versus mesenchymal cells, and the expression of a panel of cancer related genes. These were correlated to survival.
I found the study compelling in their analysis of the circulating cells captured and the spheres they created. However, the main issue is with the clinical correlation and the samples they utilized. The group seems to be quite heterogeneous. It isn’t quite clear, but it seems they indicated this group was palliative or unresectable, yet 1 patient was stage III, and a few others received chemotherapy followed by surgery. It isn’t clear either how many of the oropharyngeal cancer group were HPV+, which also would greatly affect their survival. In other words, to really categorize the group, one must perform a multivariate analysis to filter out these confounding factors.
Additionally, it is not clear how they happened to study the genes they did, and again, there are multiple comparisons being made to a heterogeneous clinical group. Not only are there too many comparisons that would warrant an adjustment for multiple comparisons, but there are also multivariable concerns that would need to be addressed.
Overall, there is potential in this manuscript, but at the moment, as the focus is on clinical correlations, there are just too many variables unaccounted for to warrant publication.
Author Response
Response to reviewer 2’s comments
This is a very interesting article, looking at the culture and isolation of circulating tumor cells in a head and neck cancer population. The authors look at the growth of the cells, the relative frequency of epithelial versus mesenchymal cells, and the expression of a panel of cancer related genes. These were correlated to survival.
Reviewer’s comment 1: I found the study compelling in their analysis of the circulating cells captured and the spheres they created. However, the main issue is with the clinical correlation and the samples they utilized. The group seems to be quite heterogeneous (1). It isn’t quite clear, but it seems they indicated this group was palliative or unresectable, yet 1 patient was stage III, and a few others received chemotherapy followed by surgery (2). It isn’t clear either how many of the oropharyngeal cancer group were HPV+, which also would greatly affect their survival (3). In other words, to really categorize the group, one must perform a multivariate analysis to filter out these confounding factors (4).
Response 1: We appreciate greatly for the reviewer’s brilliant comments. Here, we will reply to the valuable comments one by one.
(1) Yes, the population of enrollment seemed to be heterogeneous, because of two major reasons. First, HNSCC remains a tumor with heterogeneity even if the primary site of enrolled patients is the same. The analysis was based on the Intention-to-treat principle. Therefore, all patients who meet the set eligibility criteria were included for final analysis. In the present study, one of the criteria of enrollment was patients with initially unresectable HNSCC. That means that the patients were a group with unresectable status at diagnosis, who never exposed to any anti-cancer therapies. The criterion was hypothesized to be essential in CTC studies because we designed to investigate patients with HNSCC and no prior treatment alterations. The CTC counts in patients before chemotherapy will be possibly much different from CTC counts after chemotherapy. That is why we enrolled patients following a very simple rule of “no prior treatment,” which should be deemed as a same patient population although the following treatment (surgery or not) might greatly vary according to the responses to the first chemotherapy or chemoradiotherapy. As we described in the manuscript (page 8, lines 325-327), two patients were excluded from the Cox regression analysis because salvage surgeries after CCRT-resulted tumor regression were carried out, which might have greatly altered PFS in this population, just as the reviewer’s worries and comments. Second, the current study was a proof-of-concept study, which aimed to integrate a three-dimensional spheroid culture model in a CTC isolation and purification process. Our purpose of the investigation was technological advances. Therefore, we report the exciting findings in a relatively small population with the same criteria of unresectability in patients with newly diagnosed HNSCC. We hope that the reviewer could consider the different design of the current investigation compared to conventional clinical studies.
(2) Regarding one patient with stage III oropharyngeal cancer was included in the study, the unresectability was determined because of (i) organ preservation purpose and (ii) medically unfit for surgery, considering pre-existing medical comorbidities.
(3) The prevalence of HPV-positive HNSCC in Taiwan is not such high as that in Western countries (Laryngoscope, 2016, 126:1097; Oral Oncology, 2008, 44:174; IJC, 2015, 137:395). The HPV test is not routinely performed in patients with non-oropharyngeal cancer. In response to the reviewer’s query about p16 status in the current study, we have five of nine (55.6%) p16-positive oropharyngeal cancer patients. Because the HPV test results were available in only 25% of total cases, p16 positivity was not chosen as a covariant in Cox regression analysis. If we added p16 status (positive versus negative or unknown) into the univariate analysis for PFS, the results will be similar.
Figure RR1 The univariate analysis of p16 status to progression-free status. (p16-positive versus p16-negative/p16-unknown). |
(4) We greatly thank the reviewer for the valuable comments of multivariate analysis. The multivariate analysis under LR forward model has been done, and all the covariates (age, gender, tumor site, ECOG performance status, treatment modality, and CTC counts) in univariate analysis were sent into multivariate analysis. The results showed that M-type CTC counts were the only independent prognostic factor for PFS after adjusted all other covariates (P=0.029, HR=1.153, 95%CI=1.015-1.310). The description has been added to the Result and discussion section. (please refer to the manuscript of Highlighted Revision; page 9, lines 344-348)
Reviewer’s comment 2: Additionally, it is not clear how they happened to study the genes they did, and again, there are multiple comparisons being made to a heterogeneous clinical group. Not only are there too many comparisons that would warrant an adjustment for multiple comparisons, but there are also multivariable concerns that would need to be addressed.
Response 2: Thanks for the reviewer’s suggestion. The genes we studied in current work were those who are critical in multiple drug resistance, EMT process, and acquiring stemness. These three programs were thought closely related to tumor metastasis and relapse (Nat Rev Cancer, 2002, 2:48 and 2009, 9:265; Nat Rev Clin Oncol, 2017, 14:611). This is the reason why we chose these genes for further study. The expressions of 17 genes were investigated in this study. Among them, one gene (CDH2) with low detection rate in our study population was excluded in the log-rank survival analysis. To deal with the probability of multiple tests, the significance P-value in the gene study was set at 0.003 by the Bonferroni correction (0.05/n). The description has been added to the material and method section. (please refer to the manuscript of Highlighted Revision; page 5, lines 229-230)
Reviewer’s comment 3: Overall, there is potential in this manuscript, but at the moment, as the focus is on clinical correlations, there are just too many variables unaccounted for to warrant publication.
Response 3: We much thank the reviewer for the comments. We realize that the doubt raised from the reviewers would be the small sample size in the current proof-of-concept study and fully understand that a conclusion should be made with cautions. However, we are still excited about the technologic advances in the current investigation and hope to share the updated results with the researchers in this field. Further large-scale studies on specific genes of interests from this study will be doable and eagerly warranted.

Reviewer 3 Report
The Manuscript submitted by Liao et al. entitled:" The integration of a three-dimensional spheroid cell culture operation
in a circulating tumor cell (CTC) isolation and purification process:
The clinical significance and prognostic role of the CTCs isolated
from the blood samples of head and neck cancer patients" focusses on a very imprtant and highlighted topic regarding actual limitations of positive and negative selection-based circulating tumor cell (CTC) isolation methods, which are unable to resolve CTCs that undergone EMT and represent the most agressive metastasis - relevant cancer cells. Authors introduced a two-step methode consisting a negative selection CTC isolation and subsequent spheroid cell culture. The provided method offers two advantages: a more efficient isolation of cancer metastasis - related CTCs, and the spheroid cell culture used could serve as a cell culture model mimicking the process of new tumor tissue formation during cancer metastasis.
Authors provide appropriate introduction to the current CTC isolation techniques, their limitations, as well as to three-dimensional (3-D) spheroid cell culture.
The submitted work is appropriate the data and their presentation are of good quality, a minor comment is provided to the Methods:
Comment
Page 4, lines, 164 - 172. please mention th manufacturer of the 48 well cell culture microplates, and if these are low adhesion plates.
Author Response
Reviewer’s comment 1: The Manuscript submitted by Liao et al. entitled:" The integration of a three-dimensional spheroid cell culture operation in a circulating tumor cell (CTC) isolation and purification process: The clinical significance and prognostic role of the CTCs isolated from the blood samples of head and neck cancer patients" focusses on a very imprtant and highlighted topic regarding actual limitations of positive and negative selection-based circulating tumor cell (CTC) isolation methods, which are unable to resolve CTCs that undergone EMT and represent the most agressive metastasis - relevant cancer cells. Authors introduced a two-step methode consisting a negative selection CTC isolation and subsequent spheroid cell culture. The provided method offers two advantages: a more efficient isolation of cancer metastasis - related CTCs, and the spheroid cell culture used could serve as a cell culture model mimicking the process of new tumor tissue formation during cancer metastasis.
Authors provide appropriate introduction to the current CTC isolation techniques, their limitations, as well as to three-dimensional (3-D) spheroid cell culture.
The submitted work is appropriate the data and their presentation are of good quality, a minor comment is provided to the Methods:
Response 1: Reviewer’s positive comments are appreciated.
Reviewer’s comment 2: Page 4, lines, 164 - 172. please mention the manufacturer of the 48 well cell culture microplates, and if these are low adhesion plates.
Response 2: The 48-well microplates used in this study were the regular cell culture microplates. In this study, the regular microplates were pre-coated with 2% agarose at the bottom of each well for achieving low adhesion purpose. These information as well as the information of manufacturers has now been supplemented in the manuscript (Please refer to the manuscript of Highlighted Revision; Page 4, Line 180-182).

Reviewer 4 Report
Major point
This study has too little data to discuss 'Circulating small cells'.
Therefore, I require Major Revision.
・Please describe the analysis method of this study in detail in material and methods section.
・First of all, the definition of CTC should be clearly understood.
・There are too few case number for academic arguments. In this study, please respond by increasing the number of cases.
・This paper has not been analyzed and discussion of the problem.
・And, please describe the discussion of the mechanism.
Mainor point
・Please list the technique about the statistical analysis properly.
・The sentence of this paper has many careful mention errors. Please review it.
Author Response
Reviewer’s comment 1: This study has too little data to discuss 'Circulating small cells'.
Therefore, I require Major Revision.
Response 1: We thank for reviewer’s comment.
Reviewer’s comment 2: Please describe the analysis method of this study in detail in material and methods section.
Response 2: The analytical methods used in this study have been properly described and discussed in the “material and method” section. In response to the reviewer’s comment, some other detail information relevant to analytical methods has now been newly supplemented in the “material and method” section (Please refer to the manuscript of Highlighted Revision; Page 4, Lines 167-175 and Lines 180-182; Page 5, Lines 193-195, Lines 202-210, Line 217, and Lines 234-238).
Reviewer’s comment 3: First of all, the definition of CTC should be clearly understood.
Response 3: Reviewer’s suggestion is appreciated. Conventional biochemical-based schemes for CTC isolation/purification commonly make use of specific cell surface markers, such as epithelial cell adhesion molecule (EpCAM) and cytokeratins (CKs) to define CTCs. Namely, these CTC isolation techniques use these two surface markers to differentiate the CTCs from the surrounding blood cells. These information has been provided in the manuscript. In order to make the definition of CTC clearer, the original description has now been modified (Please refer to the manuscript of Highlighted Revision; Page 2, Lines 58-60).
In addition to the conventional method for the definition of CTCs as abovementioned, this study also defined the “E-type CTCs”, and “M-type CTCs” as the CD45∪CD11bnegative (neg)/EpCAM∪CK19positive (pos) nucleated cells, and the CD45∪CD11bneg/VIMpos nucleated cells, respectively. These information has been provided in the “Results and discussion” section (Please refer to the manuscript of Highlighted Revision; Page 6, Lines 282-284).
Reviewer’s comment 4: There are too few case number for academic arguments. In this study, please respond by increasing the number of cases.
Response 4: Many thanks for the reviewer’s comments. We agree that small sample size is the weakness of this study. In this work, we aimed to initially explore the clinical significances of the CTCs isolated via the 2-step CTC isolation method. The preliminary results obtained from this study will be valuable for scientists to design a new study to further explore the issues relevant to this work. Of course, further studies with larger sample size are worthy to conduct. The above descriptions have been described in the “Introduction”, and “Conclusions” section (Please refer to the manuscript of Highlighted Revision; Page 1, Lines 4-5 and Line 35; Page 3, Lines 126-127; Page 8, Lines 314 and 349; Page 12, Lines 443-444; Page 13, Lines 473 and Lines 475-476). Reviewer’s understanding is greatly appreciated.
Reviewer’s comment 5: This paper has not been analyzed and discussion of the problem.
Response 5: Many thanks for the comment indeed. If the “problem” raised by the reviewer means the technical problem that this study would like to address, this issue has been described and discussed in the manuscript [Please refer to the manuscript of Highlighted Revision; Page 1, Line 22-25(Abstract section); Page 2, Line 53-67, Lines 75-82, and Line 83-94 (Introduction section); Page 6, Line 250-263 (Results and discussion section); Page 12, Line 448-462 (Conclusions section)].
Reviewer’s comment 6: And, please describe the discussion of the mechanism.
Response 6: Many thanks for the comment. If the mechanism pointed out by the reviewer means the mechanism of using the 2-step CTC isolation and purification method (i.e., a conventional negative selection-based CTC isolation method+ a subsequent 8-day spheroid cell culture) (1) for the isolation of CTCs, as well as the working mechanism of utilizing this approach (2) for the isolation of metastasis-relevant CTCs and for the evaluation of the potential of cancer metastasis (and thus the prognosis of disease), these information was described in the manuscript. For the working mechanism of using the 2-step CTC isolation and purification method for (1) the isolation of CTCs, it was described in the “Introduction” section (Please refer to the manuscript of Highlighted Revision; Page 2, Lines 83-87 and Lines 90-94, Page 3, Lines 95-103). For the working mechanism of utilizing this approach for (2) the isolation of metastasis-relevant CTCs and for the evaluation of the potential of cancer metastasis, it was described and discussed in the “Introduction” section (Please refer to the manuscript of Highlighted Revision; Page 3, Lines 104-114), “Results and discussion” section (Please refer to the manuscript of Highlighted Revision; Page 6, Lines 259-268), and “Conclusions” section (Please refer to the manuscript of Highlighted Revision; Page 13, Lines 445-453). We thank for the comment again. In order to highlight the working mechanisms used in this study, some of the descriptions abovementioned have now been improved (Please refer to the manuscript of Highlighted Revision; Page 2, Line 94, Page 3, Lines 98-99 and Lines 107-108)
Mainor point
Reviewer’s comment 7: Please list the technique about the statistical analysis properly.
Response 7: Reviewer’s suggestion is thankful. The technique of statistical analysis used in this study has now been described in more detail in the manuscript (Please refer to the manuscript of Highlighted Revision; Page 5, Lines 234-238).
Reviewer’s comment 8: The sentence of this paper has many careful mention errors. Please review it.
Response 8: Many thanks for reminding us this issue indeed. The manuscript had been edited by a native English speaker. Please refer to the certificate below.
Link: https://drive.google.com/open?id=1iKswWYk1AwJlWrTGy1iHTDfL_9mXO7H_

Round 2
Reviewer 1 Report
None
Author Response
We thank to the reviewer’s comments and agreement.
Reviewer 2 Report
The authors have basically responded to many of the comments but have not really fixed them as it pertains to the clinical correlations. In order to suggest that a test might have prognostic significance, once must be quite rigorous in the methodology.
The authors indicate that p16 testing is not routinely done, which is appropriate in non-oropharyngeal sites. However, 5/9 of their oropharyngeal cancer patients did have p16 positivity that they did not include in the analysis. This suggests that these patients have HPV-mediated disease and would therefore expect improved survival for that group. Additionally, they would have to be staged with the AJCC 8th edition staging which would certainly change their analyses. This is one of the reasons why with such a small cohort, it is very challenging to look at prognostic significance. Things like smoking status were also not included.
Therefore, while the study is of interest, I think that suggesting that there would be prognostic significance in their findings is too premature. They would need to focus on a particular subgroup, or look into more of the mechanistic aspects of the relevant genes in order to provide a more convincing study.
Author Response
Reviewer’s comment 1: The authors have basically responded to many of the comments but have not really fixed them as it pertains to the clinical correlations. In order to suggest that a test might have prognostic significance, once must be quite rigorous in the methodology.
Response 1: Thanks for the reviewer’s comments again. We are sorry that the first round responses and revisions did not satisfy the reviewer’s expectancy. We really cherish the opportunity of revision and also take the reviewer’s valuable comments seriously. Please excuse us explain for the reasons why the authors still wish and suggest Cancers to publish this report considering the most significant advance of this study is the methodology (two-step CTC purification for quantitative and qualitative analyses), not survival. In this work, we aimed to initially explore the clinical significances of the CTCs isolated via the 2-step CTC isolation method. As the reviewer’s comments, the authors agree with the survival impacts are just for demonstration here considering the small sample size. The title and article have now been described mor cautiously. (Please refer to the manuscript of Highlighted Revision; Page 1, Lines 4-5 and Line 35; Page 3, Lines 126-127; Page 8, Line 314 and Line 349; Page 12, Lines 443-444; Page 13, Line 473 and Lines 475-476)
Reviewer’s comment 2: The authors indicate that p16 testing is not routinely done, which is appropriate in non-oropharyngeal sites. However, 5/9 of their oropharyngeal cancer patients did have p16 positivity that they did not include in the analysis. This suggests that these patients have HPV-mediated disease and would therefore expect improved survival for that group. Additionally, they would have to be staged with the AJCC 8th edition staging which would certainly change their analyses. This is one of the reasons why with such a small cohort, it is very challenging to look at prognostic significance. Things like smoking status were also not included.
Response 2: As the reviewer mentioned, HPV-associated oropharyngeal cancer is recognized as a distinct population compared to non-oropharyngeal cancers in AJCC 8th edition. The authors heartfully agree with the clinical comments from the reviewer. In the previous revision, the authors felt sorry about misunderstanding the question of the reviewer. In response to the reviewer’s comment, the patients were therefore rechecked and confirmed that the staging (AJCC 8th edition, including p16 status) are correct in the revised manuscript. In the revised analysis, either the revised tumor stages or p16 status was deemed as covariates in the Cox regression model. The results showed that the count of M-CTC (mesenchymal-type CTC) was still an independent factor for PFS after standard adjustment by the possible confounding factors. The authors have added this information in the “results and discussion” section in the revised manuscript. (Please refer to the manuscript of Highlighted Revision; Page 9, Line 360 and Lines 369-370; Page 10, Lines 375-376 and Lines 380-381)
In addition to smoking status mentioned by the reviewer, betel quid chewing and alcohol drinking are also crucial for HNSCC carcinogeneses. However, the authors decided not to use these factors in the Cox model, considering the fact of relatively small sample size. In the case of a large-scale prospective study which the team is currently running, the authors will fully assess these factors (tobacco smoking, betel quid chewing, and alcohol drinking) based on large sample size.
Back to the present study, which is a proof-of-concept investigation, the authors would like to emphasis on the novelty, efficacy, and potential utilities of the 2-step CTC isolation and purification method in clinical scenarios. The study team has preliminarily demonstrated that the information from the cells purified with 2-step CTC isolation method might have prognostic potential in HNSCC.
Reviewer’s comment 3: Therefore, while the study is of interest, I think that suggesting that there would be prognostic significance in their findings is too premature. They would need to focus on a particular subgroup, or look into more of the mechanistic aspects of the relevant genes in order to provide a more convincing study.
Response 3: Many thanks for the reviewer’s comments. We agree that small sample size is the weakness of this study. In this work, we aimed to initially explore the clinical significances of the CTCs isolated via the 2-step CTC isolation method. To our knowledge, this is the first study bringing the concept of 3D-spheroid culture into the CTC isolation procedures and applying to clinical study. The preliminary results obtained from this study will be valuable for scientists to design a new study to explore the issues relevant to this work further. Of course, further studies with larger sample size are worthy of conducting. The above descriptions have been described in the “Introduction,” and “Conclusions” section (Please refer to the manuscript of Highlighted Revision; Page 1, Lines 4-5 and Line 35; Page 3, Lines 126-127; Page 8, Line 314 and Line 349; Page 12, Lines 443-444; Page 13, Line 473 and Lines 475-476). Reviewer’s understanding is greatly appreciated.

Reviewer 4 Report
This paper has been improved. It seems that it is possible to accept.
Round 3
Reviewer 2 Report
I appreciate the authors' edits. This should be publishable, but does require some minor proofreading.